# SARS-CoV-2 Omicron Subvariants Balance Host Cell Membrane, Receptor, and Antibody Docking via an Overlapping Target Site

**DOI:** 10.3390/v15020447

**Published:** 2023-02-06

**Authors:** Michael Overduin, Rakesh K. Bhat, Troy A. Kervin

**Affiliations:** Department of Biochemistry, University of Alberta, Edmonton, AB T6G 2H7, Canada

**Keywords:** coronavirus, SARS-CoV-2, membrane docking, lipid bilayer, spike protein, variants, omicron, delta, MODA

## Abstract

Variants of severe acute respiratory syndrome coronavirus 2 (SARS-CoV-2) are emerging rapidly and offer surfaces that are optimized for recognition of host cell membranes while also evading antibodies arising from vaccinations and previous infections. Host cell infection is a multi-step process in which spike heads engage lipid bilayers and one or more angiotensin-converting enzyme 2 (ACE-2) receptors. Here, the membrane binding surfaces of Omicron subvariants are compared using cryo-electron microscopy (cEM) structures of spike trimers from BA.2, BA.2.12.1, BA.2.13, BA.2.75, BA.3, BA.4, and BA.5 viruses. Despite significant differences around mutated sites, they all maintain strong membrane binding propensities that first appeared in BA.1. Both their closed and open states retain elevated membrane docking capacities, although the presence of more closed than open states diminishes opportunities to bind receptors while enhancing membrane engagement. The electrostatic dipoles are generally conserved. However, the BA.2.75 spike dipole is compromised, and its ACE-2 affinity is increased, and BA.3 exhibits the opposite pattern. We propose that balancing the functional imperatives of a stable, readily cleavable spike that engages both lipid bilayers and receptors while avoiding host defenses underlies betacoronavirus evolution. This provides predictive criteria for rationalizing future pandemic waves and COVID-19 transmissibility while illuminating critical sites and strategies for simultaneously combating multiple variants.

## 1. Introduction

As we enter the fourth year of the COVID-19 pandemic, an ever-larger number of SARS-CoV-2 variants are appearing. Understanding the mechanistic drivers of their divergence and transmission could inform the development of pan-therapeutic antibodies, vaccines, and inhibitors [1]. The Omicron BA.1 variant of concern exhibits exceptional transmission rates and faster replication in bronchi [2] than earlier variants that tend to enter host lung cells via fusion with plasma membranes, compromising respiratory systems before spreading into other organs, such as the brain, heart, kidney, and liver [3]. The BA.2, BA.2.75, and BA.4/5 variants have acquired mutations that evade neutralizing antibodies and affect cell–cell fusion capacity and lung cell entry efficiency [4]. The pathway of host entry involves viral particles contacting and penetrating cell membranes via cooperative sets of spike (S) proteins. Following expression in the endoplasmic reticulum (ER), new spike proteins are packaged into exocytic vesicles to form virions that leave one host cell to infect another. It is unclear how these processes are impacted by the many mutations appearing in this polyfunctional protein. 

The mutations arising in Omicron subvariants are concentrated in the head of the spike protein (Table 1). The BA.2 variant arose in Denmark, India, and South Africa in early 2022, and it compromises the neutralizing activity of almost all therapeutic antibodies having gained T376A, D405N, and R408S mutations and lost the G446S and G496S mutations found in BA.1 [5]. The subsequent BA.2.12.1 variant gains a L452Q mutation in the spike head, and after emerging in the US in early 2022, became the dominant variant there [6], while the BA.2.13 subvariant contains a similar L452M mutation. Few BA.3 variant sequences appeared and differ at only three positions from both BA.1 and BA.2. The BA.4 and BA.5 variants possess identical spike sequences with the same mutations over BA.2 and are referred to as BA.4/5 here. They spread rapidly around the world in the summer of 2022, with mutations of L452, F486, and R493 conferring resistance to antibodies that target the RBD, supporting the notion that immune system evasion is now a key selective pressure [6,7]. The BA.2.75 variant evolved from BA.2 in India, where it is spreading rapidly, and it displays growth advantages over BA.5 [8]. More recent variants that evade antibodies are spreading rapidly including BQ.1, which arose in Africa and contain spike mutations K444T and N460K, as well as XBB, which may have emerged in South Asia and gained spike mutations V455P and N460K [9]. The increased number of highly transmissible variants underscores the need to identify the key sites and drivers for their evolution to develop strategies for robust intervention. With each season, there is a risk that another generation of antibody-based therapeutics will become obsolete, suggesting that therapeutic strategies should focus on critical sites to offer the broadest protection. Most important for spike function is the surface mediating host cell binding and entry, as well as vesicle docking during virion packaging and trafficking.

The structure and function of the spike (S) protein provide a basis for understanding the effects of variant mutations. It comprises S1 and S2 subunits, which are responsible for host cell recognition and membrane fusion, respectively. The spike protein forms a trimer, which amplifies the effects of mutations three-fold. The spike head, where most Omicron mutations are found, is composed of a N-terminal domain (NTD) and receptor binding domain (RBD), both of which present membrane binding surfaces in their closed and open states [10,11]. Spike opening involves raising of the RBD, whereupon it can recognize an ACE-2 molecule on a host cell through the receptor binding motif (RBM). The S protein also contains fusion peptide (FP) and heptapeptide repeat (HR) sequences, a transmembrane helix, and a short cytoplasmic region that is heavily palmitoylated. Despite thousands of structures of spike subunits being available [12] including from Omicron subvariants (Table 2), all lack resolved lipid bilayers, necessitating computational methods to assess how they mediate membrane interactions. 

The membrane docking sites of spike trimers can be predicted by tools, including DREAMM [29,30], Ez-3D [31], positioning of proteins in membranes (PPM) [32,33], and the membrane optimal docking area (MODA) program [34]. The latter is trained to identify lipid binding surfaces and calculates membrane binding propensities for each residue in a protein structure. We previously used MODA to discover membrane binding sites in bacterial and viral trafficking proteins [35,36], eukaryotic membrane readers [37,38], and spike proteins [10,11]. Here, we use this approach to reveal how Omicron subvariant mutations affect the spike’s abilities to attract and engage membranes. We relate this to spike conformational dynamics, ACE-2 receptor affinity, and antibody evasion, showing how these forces are balanced and illuminate the design of pantropic therapeutic agents.

## 2. Materials and Methods

### 2.1. Spike Sequence Analysis

The SARS-CoV-2 spike protein sequences were obtained from UniProt entry P0DTC2 [39], and mutations were also identified from the literature, as were sites of binding to the ACE-2 receptor and antibodies [5,6,7,27,28,40].

### 2.2. Structures of Spike Proteins

Structures of the available SARS-CoV-2 Omicron variant spike trimers, including ACE-2 complexes, were obtained from CoV3D [12] and RCSB Protein DataBank (PDB) [41]. Structures of trimers with resolutions of typically at least 3.5 Å were prioritized (Table 1), as they better define the sidechain conformations and surface residue positions. Multiple structures of the same conformational state of a given spike ectodomain were compared (if available) to improve confidence in MODA scores. The ratios of the up and down states of the RBD modules, as well as complexes with ACE-2 receptor molecules, were contrasted based on their reported populations in cEM structures.

### 2.3. Membrane Binding Sites

The membrane binding propensities of each residue in structures of Omicron spike trimers were predicted using MODA, which generates a score ranging from 0 up to ~3600 based on features known to bind lipid bilayers [34]. The sites were depicted with the ICM Browser [42], with missing coordinates remaining unmodelled, and similar structures were considered to address such gaps. The aggregate MODA score of the RBD was calculated, as the RBD is typically the most resolved, exhibits the largest membrane binding propensity, and is the most exposed viral appendage. This value includes residues corresponding to L335, G339, N343, T345, V367, N370-S375, N437, N439, N440, K444-G447, Y449, N450, L455, F456, T470, I472, A475-P479, N481-F490, L492-S494, F496, and Q498-Q506 of the wild-type SARS-CoV-2 sequence. For cross-validation of the membrane binding residues identified by MODA we used the DREAMM [29,30] and PPM3 programs [32,33], the latter with a planar plasma membrane simulation and extracellular N-terminus. Further confirmation of membrane attraction was obtained from the dipole moments of closed spike trimer structures using the Protein Dipole Moments Server (https://dipole.proteopedia.org, accessed on 15 December 2022) [43].

### 2.4. Statistical Analysis

Membrane binding propensities were considered significant if at least two residues in sequence or structure exhibited MODA scores of 20 or more. Lone residues with MODA scores over 40 were considered to exhibit significant and substantial membrane binding. The output files generated by MODA for each multi-subunit structure were further analyzed in Excel (Microsoft) to determine significant differences in scores of residues and sites in the various states and to generate heatmaps to compare patterns. Any negative MODA scores were adjusted to zero, and average scores and standard deviations were calculated based on values from residues in each subunit of a trimer. Linear regression tests were performed to calculate whether the dipole moments of the spike ectodomain structures were related to reported ACE-2 affinities of their RBD modules.

## 3. Results

The membrane binding surfaces of spike structures from nascent SARS-CoV-2 variants were compared to those of the original Omicron BA.1 type. In particular, 54 Omicron spike structures determined by cryo-EM that represent various functional states (Table 2) were analyzed in order to identify and cross-validate the residues and poses involved in membrane binding. The effects of the various mutations that are gained and lost (Table 1) were assessed, revealing that the high membrane binding propensity of the ACE-2-free trimers is retained, as is ACE-2 binding affinity. Overall, the local changes in membrane binding surfaces and dynamics appear to compensate for the alterations gained in order to evade host immune systems, as described below for each new variant.

### 3.1. Conformations and Membrane Binding of BA.2 Spike Trimers

The membrane binding propensities of BA.2 spikes are similar to those of BA.1 spikes, indicating a conserved approach during viral docking to host cell membranes. The pathway of Omicron spike protein interaction with membranes (Figure 1) begins with the closed structure (state 1.0), which is in equilibrium with the open state, wherein one RBD elevates (state 1.1) based on the population of cryo-EM structures [7,26]. An ACE-2 receptor is then bound by the sole elevated RBD. Another RBD rises and is then latched to a second ACE-2 molecule. Finally, the third RBD moves up and is bound by the third ACE-2 molecule to form the symmetric receptor assembly [25]. This pathway through the network of potential spike states represents an efficient trajectory that conjoins viral and host membranes in preparation for fusion. It illustrates the multiple conformers by congregated spikes that can contribute to formation of a stable contact site, transient hemifusion intermediate, and expeditious transfer of viral materials into a host cell.

The distribution of states in the Omicron subvariants differ. While the BA.1 spike trimer preferentially adopts the open state with a single RBD up in cryo-EM samples, the closed and open conformers (states 1.0 and 1.1) are present in roughly equal proportions in BA.2 [25] or with a slight preference for the closed state [26]. This population difference could alter host cell membrane interactions. However, the membrane binding propensities of the closed BA.1 and BA.2 spikes are similar, as are those of their open states. Hence the membrane interactions of the ACE-2-free network of BA.1 states is retained in BA2. In contrast, the higher ACE-2 affinity of the BA.2 spike compared to BA.1 [7,27,28] would favor formation of the multiply ACE-2-bound states, thus stabilizing specific complexes with host cell membranes.

The effects of mutations acquired by the BA.2 spike largely balance out in terms of membrane binding propensity, with the following contributions:The T19I mutation in BA.2 adds exposed hydrophobicity and increases the membrane binding score of this position from 33.8 ± 10.8 to 214.1 in structures where this element is best resolved (PDB: 7tnw vs. 7ub0-b).The gain of the ΔL24, ΔP25, ΔP26, and A27S mutations in BA.2 increases the degree of disorder locally, although the S27 residue is resolved with significant membrane binding propensities in all the BA.2 spike:ACE-2 complexes (PDBs 7xoa, 7xob, 7xo7, 7xo8, Figure 2).The loss of the A67V, ΔH69, and ΔV70 mutations in BA2 leads to significant membrane binding propensities appearing at the H69 and V70 positions in structures where this loop is resolved (PDB: 7ub5).The loss of the T95I mutation in BA2 eliminates the membrane binding propensity that appears at this position in a minority of structures, such as the nearby K97. Still, it occasionally appears as membrane-binding in closed structures (e.g., PDB: 7ub5-a, 7ub6-c).The loss of the ΔV143, ΔY144, and ΔY145 mutations in BA2 introduces significant membrane binding propensity here in ACE-2-free structures where this element is resolved (e.g., PDB: 7ub5-a, 7ub5-ab, and 7ub6-a).The loss of the ΔN211, L212I, and ins214EPE mutations and the gain of the V213G mutation in the BA.2 NTD alters the flexibility and removes exposed hydrophobicity and charge in this mobile loop, which retains significant but variable and reduced membrane binding propensities in ACE-2-free structures.The T376A mutation in the BA.2 RTD decreased the membrane binding propensity of the neighbouring F375 residue from 67.2 ± 21.8 (PDB: 7tnw) to zero in all the BA.2 closed structures and induced conformational changes in this loop that reduces antibody interactions [7].The D405N and R408S mutations gained in BA.2 eliminate a negative and positive charge exposed in the RBM and balance the overall dipole moment. The former mutation is particularly critical for immune evasion [7], while the latter alters a binding pocket for fatty acid molecules of interest for its potential druggability [45,46,47].The loss of the G446S and G496S mutations found in BA.1 eliminates glycine residues that are common in membrane binding sites and allow flexibility during membrane insertion, although significant membrane binding propensities are retained around these positions.The D796Y mutation in BA.2 results in significant membrane binding propensity appearing in Y796 and I794 in approximately half of the structures.

For confirmation of the MODA results, PPM3 was used to predict which BA.2 spike residues are most likely to interact with a lipid bilayer slab. PPM3 indicates that the following residues within the various ACE-2-free BA.2 trimer structures are likely to bind membranes:7xix: all three V445 in chains A, B, and C,7ub0: V16, N122, T124, F133, F140, L141 (chain B), and F486 (chain C),7ub5: V483-F486 (chain A),7ub6: F133, Q134, F140, L141 (chain B), V483, F486 (chain C), and7xiw W152, F486 (chain D).

All of these 22 residues, except T124, exhibit significant membrane binding propensities based on our MODA analysis. This represents agreement of 95.5%, thus confirming the membrane binding surfaces in the various BA.2 structures overall. The DREAMM program was also used to validate the BA.2 spike membrane docking sites predicted by MODA. This concurs that F152, G258, F483, Y486, and Y796 in all three subunits are membrane interactive in the highest resolution structure (PDB: 7xix). Unlike MODA, residues Y144 and L249 in the B chain and Y144 in the C chain are predicted to be membrane interactive in this structure by DREAMM. However, MODA does find these residues to be membrane interactive in other BA.2 closed structures (PDBs: 7ub0, 7ub5 and 7ub6), thus providing general agreement in terms of the sites of membrane insertion.

Further validation was obtained by comparison with the electrostatic dipoles of the spike trimers, which would provide attraction to membrane surfaces. The BA.2 spike trimer exhibits a preference for concave membranes based on a ΔG_transfer_ of −12.0 kcal/mol (vs. −7.5 kcal/mol for a planar membrane) for the highest resolution structure (PDB: 7xix). The dipole moments are strong and consistent, at 10,040 ± 190 debyes for the four available structures of its closed state (Table 2), values that are similar to that of closed Omicron BA.1 spike trimers (10,156 ± 802 debyes for PDBs 7wp9, 7wk2, 7tnw, 7tf8, and 7tl1). The open states exhibit a similar pattern, with a BA.2 spike trimer dipole of 6354 debyes being in the range of those of BA.1 spikes. Thus, both the overall membrane binding propensity and electrostatic dipole of BA.1 spikes was retained in BA.2, supporting a conserved binding mode for host cells.

### 3.2. Conformations and Membrane Binding of BA.2.75 Spike Trimers

Seven cryo-EM structures of the spike trimer from the BA.2.75 subvariant are available [27,28] (Table 2) and display overall membrane binding profiles that are similar to BA.1 and BA.2. The trajectory of host cell membrane binding states follows that of BA.2 (Figure 1), although there is a preference for the RBD-up position seen at pH 5.5, thus favouring endosomal entry [28]. The overall membrane binding propensities of the BA.2.75 RBDs are similar to those of BA.2. The dipole moment of the closed BA.2.75 spike trimer is 8981 ± 721 debyes, which is lower than that of BA.2 (10,040 ± 191, Table 2). The BA.2.75 RBD displays significantly higher affinity for ACE-2 receptors than other subvariants [27,28]. Hence, the higher apparent fusogenicity and pathogenicity of BA.2.75 [27] may be attributable to enhanced ACE-2 affinity along with retention of membrane docking, but they are tempered by reduced attraction due to its lower dipole moment. The nine mutations acquired by the BA.2.75 spike over BA.2 impact membrane binding in the following ways:The gain of the K147E mutation in BA.2.75 removes a positive charge from an exposed, dynamic loop, and membrane binding propensity position, which are present here, e.g., in BA.1 (7tof) and BA.2 (7xiw)The gain of the W152R mutation in BA.2.75 maintains membrane binding propensity at this NTD position in open and ACE-2-bound structures (PDBs: 7ypt, 7yqv, 7yr2, 3yr2), while in the closed structures, this element is disordered. However, the Trp at this position in BA.2 structures scores higher by MODA than the Arg here. The gain of a positive charge here compensates for the charge reversal at residue 147 that contributes to the dipole moment.The gain of the F157L mutation maintains membrane binding propensity at this NTD position in one subunit of BA.2.75 structures (PDBs: 7yqv, 7yr2). This residue is packed against Q14 and R158 (both of which are predicted to be membrane interactive by MODA) while being located at a C-terminus near a dynamic loop that is disordered in closed states. The Phe at this position scores higher by MODA in the BA.2 open structure (PDB: 7xiw), inferring greater membrane interactivity.The gain of the I210V mutation increases the membrane binding propensity at this NTD position in BA.2.75 structures (all but PDB 8gs6), with adjacent Pro^209^ and Leu^212^ often also being membrane interactive by MODA analysis.The gain of the G257S mutation in BA.2.75 maintains membrane binding propensity at this NTD position (as do neighbouring S256 and W258) in open and ACE-2-bound structures (PDBs: 7ypt, 7yqv, 7yr2, 3yr2), while in the closed structures this element is disordered.D339H is a novel mutation that induces local conformational changes, with this position packing against Phe371 [27]. This does not impart significant membrane binding propensity here in any BA.2.75 structure. The H339 sidechain is exposed at the top of the head, and is close to L335, which is membrane interactive (PDB: 8gs6, 7yqu, 7vqv, 7yr2). The positive charge at position 339 in low pH environments found along endocytic routes could favour membrane binding.The gain of the G446S mutation imparts resistance to antibodies [27] and maintains strong membrane binding propensity at this critical RBM position in all BA.2.75 trimers. This mutation compromises ACE-2 interactions [27], as well as flexibility that could be expected to facilitate membrane insertion.The gain of the N460K mutation in BA.2.75 contributes to the electrostatic dipole that favours membrane interactions but does not impart membrane binding propensity at this position in any BA.2.75 structures. Surprisingly, this mutation has been shown to either enhance ACE-2 affinity [27] or to have no such effect [28].R493Q maintains membrane binding propensity at this critical RBM position in a subset of BA.2.75 structures, and it also restores ACE-2 receptor affinity [6].

The various membrane binding residues in the spike trimer subvariant, including those that are mutated in BA.275, are shown in Figure 3, and they resemble the patterns seen in BA.2 (Figure 2).

For confirmation of the MODA results, we used PPM3, which predicts which residues in a structure are most likely to interact with a lipid bilayer slab. This indicates that the following residues within the various ACE-2-free BA.2 trimer structures bind membranes:7yqt: F486 in chain A and S446, V483, G485, and F486 in chain B7yqu: all three V445 in subunits A, B, and C,7yqv: F486 in chain A; V483, G485, and F486 in chain B7yqw: P793 and I794 in chain B8gs6: V483, F486 in chain A, T345, K444, and V445 in B, and F133-F135, N165 in C.

All of these 23 predicted bilayer-interacting residues, except three (P793 and I794 in 7yqw and T345 in 8gs6), exhibit significant membrane binding propensities based on MODA analysis. DREAMM also predicts that F486 and Y796 in all chains are membrane inserting, as well as Y145 and F490 in chain A and F140-L141 in chain B, all of which exhibit significant MODA scores. However, a *trans*-membrane interaction by residues 793-796 would appear unlikely based on the poses of the trimer and these may instead mediate *cis*-membrane interactions with the virus. Nonetheless, the large majority of BA.2.75 spike membrane-interacting residues identified by PPM3 and DREAMM are also predicted by MODA, providing overall confirmation.

The DREAMM program was also used for cross-validation of membrane interactions by BA.2.75 spikes. This predicts membrane insertion by F486 and Y796 in all three subunits, as well as Y145 and F490 in chain A and F140 and L141 in chain B. All of these exhibit significant membrane binding propensities by MODA, providing further validation.

### 3.3. Membrane Binding by the Closed Spikes of BA.2.12, BA.2.13, BA.3 and BA.4/5 Variants

The closed states of several additional SARS-CoV-2 Omicron variant spikes have been determined, allowing analysis of the membrane binding trends. These recent Omicron spike subvariants possess mutations of L452 that endow mild and moderate escape from neutralizing antibodies [7]. The L452Q and L452M substitutions in BA.2.12.1 and BA.2.13 spikes introduce exposed polar sidechains that do not display membrane binding propensity by MODA. However, these residues are sandwiched between L492 and Y449, which are predicted to be membrane-binding in the closed state (PDB: 7xnr and 7xns, Figure 4A). The profiles of membrane interaction across the spike sequences are similar in BA.2.12.2 and BA.2.13 (Figure 4B). The exceptions include W64, P209, R214, A475, and G476, which are membrane binding only in the former, while N122, T124, G184, N186, and Y248 are only membrane binding in the latter. Of these, the W64, N122, T124, and W248 residues do not display membrane binding in the comparable BA.2 structure (PDB: 7xix), suggesting gains in membrane binding at these positions, as well as at L492.

The BA.3 variant spike gains D405N and L371F mutations, which are not present in BA.1, neither of which is membrane interactive in the corresponding closed structure (PDB 7xiy) [7]. These mutations are present in BA.2 spikes, with F371 exhibiting membrane binding propensity in open and ACE-2-bound structures and occupying part of the biliverdin binding pocket [48], while the N405 position is least accessible to the membrane. The BA.3 spike also exhibits a reversion in the G496S mutation, which is present in BA.1. Both Ser and Gly residues in this position are similarly membrane interactive, although the dynamics imparted by a Gly are generally more favourable for membrane insertion.

The BA.4/5 variant spike gains the L452R mutation over BA.2, which provides severe escape from neutralizing antibodies [7]. This substitution also increases the dipole moment of the spike trimer, but this position is not membrane-interacting based on MODA analysis. The F486V mutation also appears to help evade neutralizing antibodies, and while this substitution retains membrane binding, it does so with reduced propensity. The deletion of H69 and V70 in the BA.3 and BA.4/5 NTDs (which also features in BA.1) compromises a putative pH switch that could favour endocytic entry [11], while the proximal G72 and N74 residues exhibit increased membrane binding propensities (Figure 4).

The membrane binding sites identified in the remaining Omicron variants (Figure 4) were cross-validated. The PPM3 program predicts that all three V445 residues in the RBMs of the BA.2.12.1 and BA.2.13 spikes (PDB: 7xns and 7xnr) are membrane interactive, consistent with MODA (Figure 4B), which further supports the perpendicular pose of the trimer bound to host cell membranes. The DREAMM program concurs that residues W152, W258, V483, and F486 in all three subunits of the BA.2.12.1 spike insert into membranes, as well as F79, W152, M153, F157, W258, V483, F486, F490, and Y501 in the BA.2.13 structure. In contrast, DREAMM uniquely predicts that Y144 and Y796 in the BA.2.12.1 spike chains insert into membranes, as well as V143, Y144, L249, and Y796 in the BA.2.13 spike chains. However, these exceptions are near membrane binding sites in other spike structures. In particular, the Y796 residue is near P793 and I794, which MODA predicts to be membrane interactive, while MODA predicts V143 and Y144 to be membrane interactive in BA.2, BA.2.75, BA.3, and BA.4/5 spike structures. The closed structure of the BA.3 spike (PDB: 7xiy) also displays consistent membrane interaction sites in PPM3 (but inexplicably failed when submitted to the DREAMM server). This predicts membrane association by V445, Y449, V483, and A484 in chain A and F140 in chain C, all of which display significant membrane docking by MODA. Thus, despite variations in the areas, there is overall agreement in the locations of membrane interaction sites in these three Omicron subvariant spike structures as assessed by the three independent methods.

Membrane insertion for BA.4/5 spike residues Y144, Y145, W152, G257, W258, V486, and F490 is predicted by both DREAMM and MODA, indicating overall agreement. However, DREAMM also uniquely predicts that M151, Y248, and Y796 insert into membranes (numbering based on WT sequence), while MODA predicts that these are adjacent to membrane binding positions or score below its significance cutoff. Plasma membrane binding is predicted by PPM3 for BA.4/5 residues N655 in chain B, as well as P793 and I794 of chain C. These residues are not predicted by MODA to be membrane-docking in the BA.4/5 spike structure (PDB 7xnq), although positions 793 and 794 score as significant bilayer binders in other spike structures where we propose they are best positioned to interact with the *cis*-membrane of the virus. When the parameters are changed to an undefined membrane slab and alternate termini orientation, PPM predicts that V483 in chain A and F140, V143, and K145 in chain B are membrane interacting, all of which score very significantly by MODA. Thus, of these thirteen residues confidently identified as membrane binders by PPM3, all exhibit significant score in MODA, thus providing confirmation of their lipid bilayer interactions.

A consensus binding preference is evident from comparison of all the Omicron subvariant spike data from our MODA, DREAMM, and PPM analyses. The RBM surface and, in particular, residues V483-F490, are the most consistent and highest scoring residues, indicating that this projection forms the primary binding site for host cell membranes. Additional membrane binding sites predicted in most Omicron variant spike structures by multiple predictive methods include K150-W152 and S256-W258, which are exposed around the circumference of the spike head. The spike trimers consistently exhibit a preference for concave membranes when analyzed by PPM3, presumably due to these extensive membrane binding surfaces presented over the convex head of spike timers.

### 3.4. Comparison of Omicron Subvariant Interactions with Antibodies, Receptors and Membranes

Common mutations that help COVID-19 strains, including Omicron, evade the immune system while retaining host receptor binding include K417N, L452R, E484K, and F486L in the RBD surface where most antibodies bind. While these mutations ensure similar or higher ACE-2 affinities of the RBD [49], only the latter two substitutions are predicted to directly insert into membranes, while the other two are next to membrane binding motifs. This suggests that enhancing ACE-2 binding may currently be more critical to viral success than further optimization of lipid bilayer interactions.

The interactions of antibodies, ACE-2 receptors, and lipid bilayers can be contrasted to shed light on their respective influences on transmissibility of Omicron sublineages, with BA.2.75 being a particularly competitive outlier. This variant is endowed with a unique antibody avoidance and much higher ACE-2 binding properties, as well as greater fusogenicity, growth efficiency, and intrinsic pathogenicity over BA.2, giving it an edge that may largely be due to acquisition of N460K and R493Q in the RBD [27,28,50]. These positions are not predicted to directly bind membrane in the respective spike trimer structures, although they are close to F456 and F490 which do. Hence, receptor binding and immune escape can be interpreted to represent primary and secondary drivers for the emergence of such mutations, which maintain host receptor and membrane binding capacities while avoiding antibody interactions.

### 3.5. Trends and Relationships of Evolutionary Drivers of Omicron Spikes

The overall membrane binding propensity of the RBD is retained in Omicron subvariants, despite the local variations due to spike mutations. The BA.1 variant spike is particularly mobile, being more prone to repositioning its RBDs, as reflected by the wider range of its MODA scores and dipole moments in structures (Table 1). In contrast, the membrane binding propensities of the BA.2 and BA.2.75 variants are less variable (Figure 5). The single structure of the BA.2.13 spike exhibits the lowest membrane binding propensity in its RBD, while the lowest dipole moment belongs to BA.2.75. In contrast, the BA.2.12.1 and BA.3 spike RBDs both exhibit high membrane binding propensities, with the BA.3 ectodomain displaying the largest dipole moment. Although beyond the scope of this work, the determination of further structures of these spike variants would allow the significance and pathway of membrane interactions to be more fully explored.

Interestingly, the dipole moments of the closed Omicron spike ectodomains that attract membranes are correlated with receptor binding. That is, the reported ACE-2 affinities [7,27,28] exhibit a positive linear relationship based on a linear regression test [F = 14.5, P = 0.0189, R^2^ = 0.784] (Figure 5D). This infers that the Omicron variant BA.2.75 has accumulated mutations that increase its ACE-2 affinity at the expense of electrostatic attraction to the membrane surface via its reduced dipole moment. Nonetheless, membrane binding propensity does not appear to have been significantly compromised in any of the Omicron variants (Figure 5A). In contrast, BA.3 has achieved a strong dipole moment but compromised receptor affinity, while retaining membrane binding propensity. The higher transmissibility of the BA.2.75 subvariant compared to BA.3, which appeared only briefly before being out-competed, suggests that ACE-2 affinity outflanks dipole moments, with membrane binding being relatively constant in these Omicron strains. Based on the various properties of the mutant spikes discussed here, the drivers for transmissibility of the Omicron variants then include:Membrane docking propensity via the lipid binding surface of the spike head.ACE-2 receptor binding affinity via the RBM of the open state of the spike trimer.Neutralization by antibodies involved in immune escape.Electrostatic membrane attraction via the dipole moment of the spike trimer.

The transmissibility (*T*) of a SARS-CoV-2 Omicron variant can be predicted as being proportional to the four “MANE” drivers, as described in the following expression:$$T\;\propto \;\frac{1}{𝒲A}\left(𝒳N\;\cdot \;𝒴M\;\cdot \;𝒵E\right)$$
where 𝒲, 𝒳, 𝒴 and 𝒵 represent the relative weightings of the *A* (ACE-2 receptor affinity), *N* (neutralization of antibodies), *M* (membrane binding propensity), and *E* (electrostatic dipole moment) terms described above. Although transmissibility is dynamic, being dependent on changing factors such as vaccination and infection rates and waning immunities, the values of the four driver terms can be measured by receptor binding assays, antibody neutralization data, and MODA and dipole moment server analysis of spike structures.

Along with structural data and the tools used here, this provides a basis for predicting the transmissibility of future SARS-CoV-2 variants. A caveat is that transmission rates also depend on other factors, such as spike protein stability and protease cleavage, viral entry pathway, replication, assembly, and egress. Hence, although the identification of key drivers in past variants can be used to rationalize effects of current mutations, future waves of COVID-19 may expose additional determinants of viral fitness.

## 4. Discussion

The evolution of betacoronaviruses has yielded a series of variants emerging with new ways to optimize host entry and propagation while avoiding immune system defenses. The mutations differentiating SARS-CoV-2 from SARS-CoV-1 provided more stable and cleavable spikes with more extensive ACE-2 receptor interactions that improved entry into host cells [51]. The subsequent variants, and especially the heavily mutated Omicron BA.1 spike, exhibit increases in the extent of membrane docking surfaces in closed and open states. Along with the higher population of the latter state, this may increase fusion-readiness and transmissibility [10]. We propose that the BA.1 lineages were out-competed when further mutations enabled escape from neutralizing antibodies originating from previous infections or vaccinations, while ACE-2 receptor and membrane binding facilities were retained. In some cases, the impact on ACE-2 affinity is not clear. For example, the BA.4/5 RBD affinity for ACE-2 is increased by three and two times over that of the BA.1 and BA.2 forms in one study [40], while another shows that the BA.3 and BA.4/5 spike constructs have lower ACE-2 affinities than those of BA.2 [7]. Nonetheless, the recent gain of almost an order of magnitude of ACE-2 binding affinity by BA.2.75 [27,28,50] is concerning and leaves considerable room for other mutations to evolve to escape immune defenses and enhance host cell membrane interactions.

The multiple steps of spike engagement with host cells provide opportunities for further mutational selection. The first opportunity is the docking to lipid bilayers of host cells by the spike head to facilitate recognition of co-located ACE-2 receptors. The spikes may also lean in and form cooperative struts against the membrane that buttress key membrane contact sites and support pre-fusion and hemifusion intermediates seen by cryo-electron tomography [44] and modelling studies [52] (Figure 1). The assembly of virions within host cells involves apparent interaction of spike heads with the surrounding envelope of vacuoles seen by electron tomography [53], as well as inside exocytic vesicles that fuse with the plasma membrane to release viral particles into the extracellular space. The transmission of virions between host cells through syncytia and filopodia involves membrane reorganizations as seen in infected cells by helium ion microscopy [54], with spikes positioned to play a major role. There are multiple lipid ligands that interact with spike trimers, as we described earlier [10,11]. Complexes are formed between spikes and neutral lipids [55], phospholipids [56], fatty acids [45,46,47], biliverdin [48], and cholesterol [57,58] that contribute to cell–cell fusion [59] and viral entry [60]. Other candidate lipid ligands include membrane raft components and gangliosides, which can be accommodated within SARS-CoV-2 NTD structures [61,62], as well as PtdIns(4,5)P_2_, which colocalizes with spikes in the plasma membrane [63]. Structures of complexes of SARS-CoV-2 spike trimers bound to lipids including fatty acids have been resolved [45,47,64] and suggest how cholesterol [57] could influence viral entry [60]. The SARS-CoV-2 RBD interacts with glycolipids, in particular those containing sialic acid [65] and heparan sulfate (HS) proteoglycans, which enhances ACE-2 binding [66]. The RBDs of Omicron sublineages appear to offer differing sites that accommodate HS with nanomolar affinities [67]. By analogy to other spike protein HS complexes [68], we infer that such membrane interactions are responsible for initial docking of the virus to the extracellular surface of host cells. Images of native spikes of SARS-CoV-2 viral particles in cells suggest that the heads also abut ER, Golgi, vesicular, and plasma membrane surfaces during intracellular assembly and trafficking [69,70,71,72,73], and we propose that similar spike sites mediate lipid interactions with such compartments. Altogether, this suggests that a diverse array of membrane interactions mediated by SARS-CoV-2 virion spike heads throughout their journey into and within host cells are impacted by spike mutations, making the relevant docking surface a complex yet attractive site for intervention.

Given the rapid rise of several highly transmissible Omicron subvariants, new capacities are likely to emerge and enhance viral fitness. A diverse range of mutations and recombinations confer multifaceted advantages for spike protein function, including enhanced host cell membrane docking, ACE-2 receptor affinity, and antibody evasion, which we propose can together predict transmissibility. The multiple capabilities utilize an overlapping surface of the RBM, which is most critical for spike function, and thus represents the key target for the design of therapeutic agents. However, the size, variability, and exposure of this surface presents challenges for specific inhibitor design, necessitating the use of multivalent boosters and multiple neutralizing antibodies. Even these defenses are soon obsolete due to resistant variants that emerge in reservoir hosts and human populations, spreading rapidly as international travel restrictions and preventative health measures relax. Blocking viral entry with pantropic inhibitors targeting this most critical spike surface, such as multivalent polymers, offers a potential avenue for combating the evolution of future variants and pandemic waves.

## Figures and Tables

**Figure 1 viruses-15-00447-f001:**
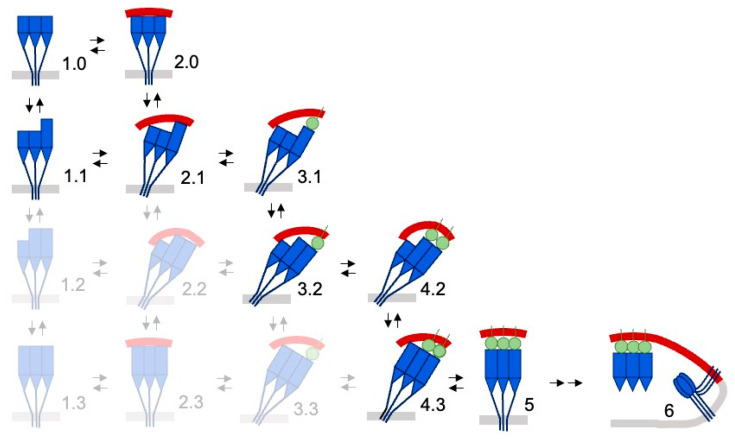
Hypothetical model of host cell membrane binding by Omicron subvariant spikes. A network of spike states based on Omicron subvariant structures is shown. Those states populated insufficiently to be seen by cryo-EM are whited out. The unbound RBD modules flicker between up and down states (tall and short blue rectangles, respectively) to yield open state 1.1. Additionally, drawn are the NTDs (blue triangles) and C-terminal remainder of the S subunits (blue line) that span the viral membrane (light grey bar). The host cell membrane (red) is engaged in states 2 and above of the spike trimer structure, which are predicted to prefer binding to concave membranes. The membrane-tethered spike trimer can bind a single ACE-2 receptor (green circle) on the host cell surface in state 3, a second ACE-2 molecule in state 4, and a third ACE-2 molecule to form prefusion state 5. This leads to proposed state 6, where ACE-2-spike complexes stabilize the membrane contact site while a cleaved spike trimer inserts into the host cell surface, drawing it close to the viral membrane to merge lipid bilayers [44]. Adapted from [11].

**Figure 2 viruses-15-00447-f002:**
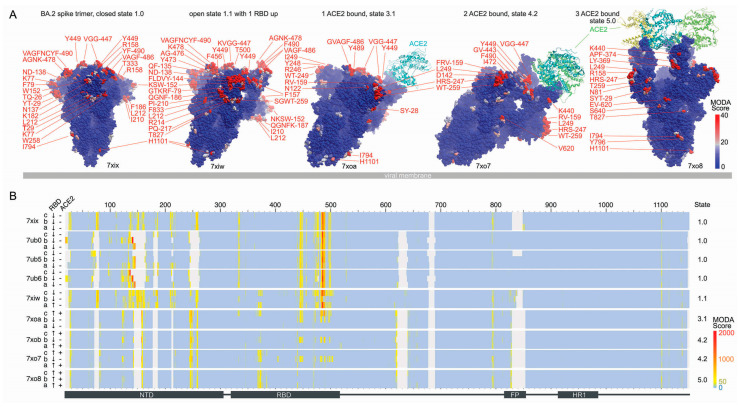
Comparison of membrane binding by BA.2 spike conformational states. (**A**) The structures of five states are shown as blue surfaces with residues having significant and substantial membrane binding propensities coloured pink and red, respectively, as indicated in the scale. The PDB entries are labeled below the structures. The viral membrane is shown as a grey slab, and the spike trimer is tilted to position the membrane binding surfaces towards the host cell above. (**B**) The heatmap shows the membrane binding propensities of residues in BA.2 spike trimer structures, with the RBD position and ACE-2 occupancy labelled on the left side. Positions are coloured light blue–yellow–red to indicate MODA scores from 0-50-2000 as in the scale, while light grey indicates missing coordinates.

**Figure 3 viruses-15-00447-f003:**
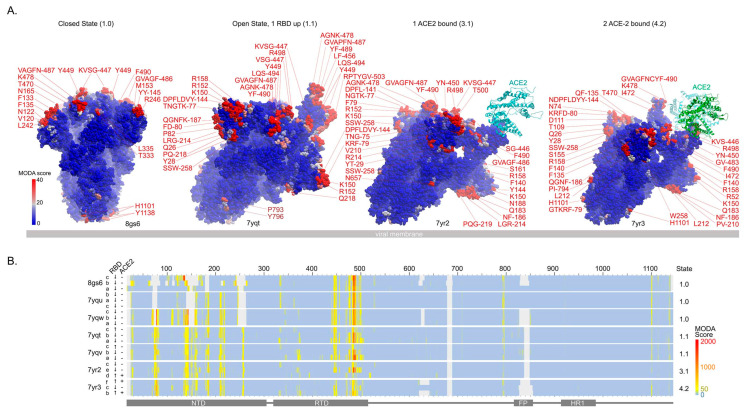
Comparison of membrane binding by BA.2.75 spike conformational states. (**A**) The residues with moderate (pink) and substantial (red) membrane binding propensities in the SARS-CoV-2 variant BA.2.75’s spike protein trimer in its progressive closed, open, and singly and doubly ACE-2 bound states (PDBs: 8gs6, 7yqt, 7yr2, 7yr3) are shown in the side views from left to right. The viral membrane is shown as a grey slab, and the spike trimer is tilted to position host membrane binding interfaces above. Residues are labelled and coloured pink-red based on MODA scores of 20–40+. (**B**) The heatmap shows the membrane binding propensities of S residues. RBD position and ACE-2 occupancy are labelled left of the map, and the S state is on the right. Positions are coloured light blue–yellow–red to indicate MODA scores from 0-50-2000, as in the lower right scale, while grey indicates missing positions in the PDB files. The positions of the NTD, RBD, fusion peptide (FP), and heptad repeat 1 (HR1) are shown below.

**Figure 4 viruses-15-00447-f004:**
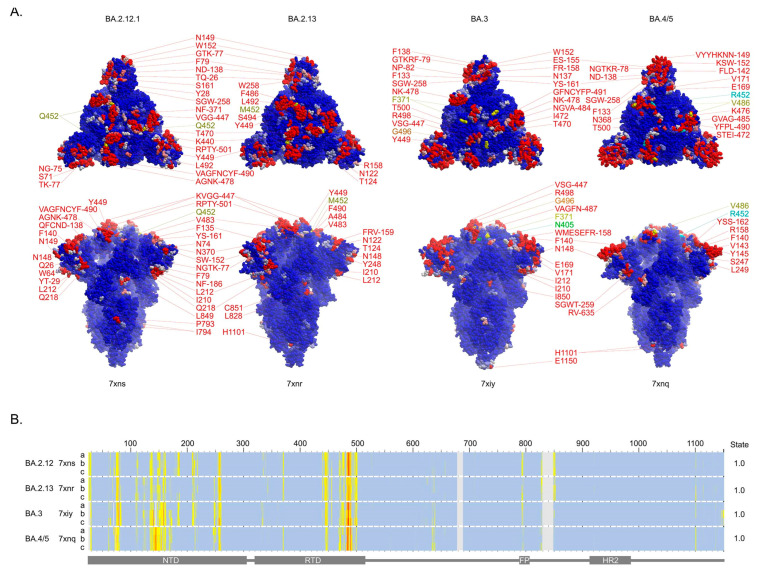
Membrane binding surfaces of closed BA.2.12.1, BA.2.13, BA.3, and BA.4/5 variants spike trimers. (**A**) The residues with moderate (pink) and substantial (red) membrane binding propensities in the closed states of spike trimer structures of the labelled variants are shown in top (**above**) and side (**below**) views of the surfaces. (**B**) Heatmap showing the membrane propensity scores for each residue of each of the three subunits of the closed spike structures, which are labelled along with the numbering of the variant sequences above and the domains beneath.

**Figure 5 viruses-15-00447-f005:**
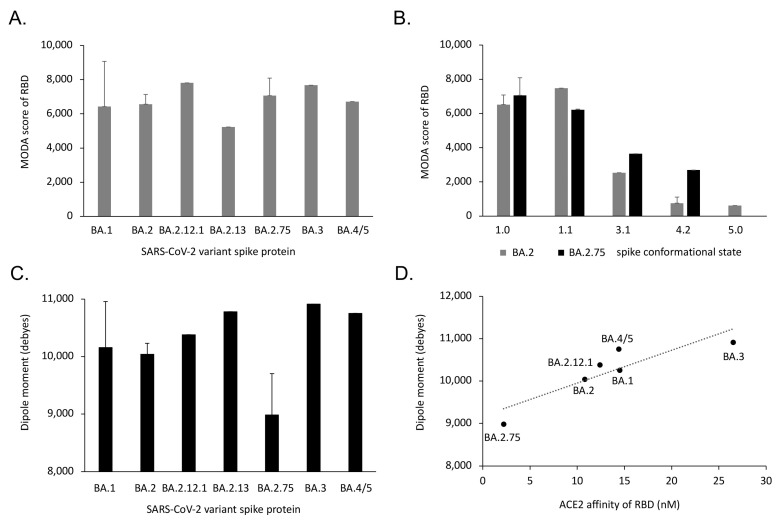
Membrane binding by conformational states of SARS-CoV-2 Omicron variants. (**A**) The average membrane binding propensities of the RBD subunit of the closed spike trimer structures are shown for each of the variants. (**B**) The average membrane binding propensities of the subunits of the closed, open, singly, doubly, and triply ACE-2-bound forms (states 1.0, 1.1, 3.2, 4.2, and 5, Figure 1) are shown for spike trimer structures (Table 2) of the Omicron BA.2 and BA.2.75 subvariants and (**C**) dipole moments of the closed structures of the Omicron subvariant spike ectodomain trimers, with standard deviations shown unless there are insufficient numbers of structures to calculate this. (**D**) The average dipole moments of the closed structures of the Omicron variant spike ectodomain trimers are linearly correlated with the ACE-2 binding affinities of their RBD domains [7,27,28].

**Table 1 viruses-15-00447-t001:** Mutations in the NTD and RBD modules of SARS-CoV-2 variant spikes emerging in 2022. Of these mutations, most are in positions predicted by MODA to bind membranes directly in multiple Omicron spike trimer structures, others are sequentially beside motifs predicted to bind membranes, and the remainder are spatially adjacent to residues that are predicted to bind membranes.

Variant	Residues That Are Mutated and Serve as Membrane Binders in Omicron Spike Trimer Structures	Next to a Membrane Binder in Sequence	Close in Space to a Membrane Binder
BA.1	ΔH69, ΔV70, G142D, ΔV143, ΔY144, ΔY145, ΔN211, L212I, ins214EPE, G339D, S371L, S373P, S375F, N440K, G446S, S477N, T478K, E484A, Q493R, G496S, Q498R, N501Y, Y505H	A67V, K417N	T95I
BA.2	T19I, ΔL24, ΔP25, ΔP26, A27S, G142D, V213G, G339D, S371F, S373P, S375F, D405N, R408S, N440K, S477N, T478K, E484A, Q493R, Q498R, N501Y, Y505H	T376A, K417N	
BA.2.12.1	T19I, ΔL24, ΔP25, ΔP26, A27S, G142D, V213G, G339D, S371F, S373P, S375F, D405N, R408S, N440K, S477N, T478K, E484A, Q493R, Q498R, N501Y, Y505H	T376A, K417N, L452Q	
BA.2.13	T19I, ΔL24, ΔP25, ΔP26, A27S, G142D, V213G, G339D, S371F, S373P, S375F, D405N, R408S, N440K, S477N, T478K, E484A, Q493R, Q498R, N501Y, Y505H	T376A, K417N, L452M	
BA.2.75	T19I, ΔL24, ΔP25, ΔP26, A27S, G142D, K147E, W152R, F157L, I210V, V213G, G257S, G339H, S371F, S373P, S375F, D405N, R408S, N440K, G446S, N460K, S477N, T478K, E484A, R493Q, Q498R, N501Y, Y505H	T376A, K417N	
BA.3	ΔH69, ΔV70, G142D, ΔV143, ΔY144, ΔY145, ΔN211, L212I, G339D, S371F, S373P, S375F, D405N, N440K, G446S, S477N, T478K, E484A, Q493R, Q498R, N501Y, Y505H	A67V, K417N	T95I
BA.4	T19I, ΔL24, ΔP25, ΔP26, A27S, ΔH69, ΔV70, G142D, V213G, G339D, S371F, S373P, S375F, D405N, R408S, N440K, S477N, T478K, E484A, F486V, Q498R, N501Y, Y505H	T376A, K417N, L452R	
BA.5	T19I, ΔL24, ΔP25, ΔP26, A27S, ΔH69, ΔV70, G142D, V213G, G339D, S371F, S373P, S375F, D405N, R408S, N440K, S477N, T478K, E484A, F486V, Q498R, N501Y, Y505H	T376A, K417N, L452R	

**Table 2 viruses-15-00447-t002:** Structures of SARS-CoV-2 Omicron subvariant spikes. All the higher resolution structures of the free ectodomain trimer and its ACE-2 complexes that are available in the PDB are listed, as are the RBD orientations, spike conformational state (Figure 1), average MODA score of the 3 RBD modules, dipole moment (Debyes) of the S ectodomain (unless “n.a.” which indicates insufficient data), numbers of bound ACE-2 molecules, resolution (Å), PDB code, and reference.

Variant	RBDs	State	MODA	Dipole	Ligand	Resolution	PDB	Reference
BA.1	3 down	1.0	n.a.	9758	-	2.56	7wp9	[13]
BA.1	down	1.0	n.a.	7853	-	2.79	7t9j	[14]
BA.1	3 down	1.0	4836	10765	-	3.10	7wk2	[15]
BA.1	3 down	1.0	3520	11247	-	3.10	7tnw	[16]
BA.1	3 down	1.0	8660	9548	-	3.36	7tf8	[17]
BA.1	3 down	1.0	8676	9460	-	3.50	7tl1	[17]
BA.1	3 down	1.0	9198	11881	-	4.00	7wg7	[18]
BA.1	1 up	1.1	9510	11663	-	3.00	7y9s	[19]
BA.1	1 up	1.1	9244	11403	-	3.00	7tgw	[20]
BA.1	1 up	1.1	5382	12246	-	3.02	7qo7	[21]
BA.1	1 up	1.1	11654	6923	-	3.11	7thk	[22]
BA.1	1 up	1.1	9475	10391	-	3.29	7tb4	[23]
BA.1	1 up	1.1	5329	11097	-	3.40	7wk3	n.a.
BA.1	1 up	1.1	9587	12937	-	3.40	7wg6	[18]
BA.1	1 up	1.1	6242	11702	-	3.40	7to4	[16]
BA.1	1 up	1.1	n.a.	n.a.	-	3.40	7tei	[17]
BA.1	1 up	1.1	6237	11215	-	3.40	7wvn	[15]
BA.1	1 up	1.1	3911	9486	-	3.41	7wvo	[15]
BA.1	1 up	1.1	9654	10175	-	3.50	7tl9	[17]
BA.1	1 up	3.1	1885	4158	1 ACE-2	2.77	7wpa	[13]
BA.1	1 up	3.1	4368	4393	1 ACE-2	2.85	7y9z	[19]
BA.1	1 up	3.1	2440	2511	1 ACE-2	2.90	7ws9	[24]
BA.1	1 up	3.1	3085	3906	1 ACE-2	3.13	7xo5	[25]
BA.1	1 up	3.1	3423	4672	1 ACE-2	3.69	7wk4	[15]
BA.1	2 up	3.2	5248	5256	1 ACE-2	3.66	7wk5	[15]
BA.1	2 up	3.2	5261	5243	1 ACE-2	3.70	7wvp	[15]
BA.1	3 up	3.3	7420	5770	1 ACE-2	4.04	7wvq	[15]
BA.1	2 up	4.2	542	n.a.	2 ACE-2	2.45	7t9k	[14]
BA.1	2 up	4.2	1070	7679	2 ACE-2	3.00	7ws8	[24]
BA.1	2 up	4.2	953	4278	2 ACE-2	3.24	7xo4	[25]
BA.1	2 up	4.2	2523	9770	2 ACE-2	3.30	7xid	[25]
BA.1	2 up	4.2	2448	8324	2 ACE-2	3.40	7xch	[19]
BA.1	2 up	4.2	1663	8544	2 ACE-2	3.50	7wgb	[18]
BA.1	3 up	5.0	112	12925	3 ACE-2	3.10	7ya0	[19]
BA.2	3 down	1.0	7252	10081	-	3.25	7xix	[7]
BA.2	3 down	1.0	6356	10173	-	3.31	7ub0	[26]
BA.2	3 down	1.0	5908	9760	-	3.35	7ub5	[26]
BA.2	3 down	1.0	6739	10146	-	3.52	7ub6	[26]
BA.2	1 up	1.1	7481	11259	-	3.62	7xiw	[7]
BA.2	1 up	3.1	2537	3265	1 ACE-2	3.20	7xoa	[25]
BA.2	2 up	4.2	511	4683	2 ACE-2	3.30	7xob	[25]
BA.2	2 up	4.2	1011	7916	2 ACE-2	3.38	7xo7	[25]
BA.2	3 up	5.0	619	12139	3 ACE-2	3.48	7xo8	[25]
BA.2.12.1	3 down	1.0	7811	10381	-	3.48	7xns	[7]
BA.2.13	3 down	1.0	5230	10781	-	3.49	7xnr	[7]
BA.2.75	3 down	1.0	8214	8158	-	2.86	8gs6	[27]
BA.2.75	3 down	1.0	6241	9500	-	3.19	7yqu	[28]
BA.2.75	3 down	1.0	6732	9285	-	3.51	7yqw	[28]
BA.2.75	1 up	1.1	6246	9661	-	3.45	7yqt	[28]
BA.2.75	1 up	1.1	6202	9646	-	3.58	7yqv	[28]
BA.2.75	1 up	3.1	3649	6056	1 ACE-2	3.30	7yr2	[28]
BA.2.75	2 up	4.2	2693	10929	2 ACE-2	3.52	7yr3	[28]
BA.3	3 down	1.0	7674	10912	-	3.07	7xiy	[7]
BA.4/5	3 down	1.0	6715	10753	-	3.52	7xnq	[7]

## Data Availability

The databases and software used here are publicly available.

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
