# Peer review of "SARS-CoV-2 Omicron Subvariants Balance Host Cell Membrane, Receptor, and Antibody Docking via an Overlapping Target Site"

_viruses, 2023, doi:10.3390/v15020447_

Round 1

Reviewer 1 Report

In this study, Overduin et al. compared the membrane binding surfaces of different Omicron subvariants by using cryo-electron microscopy and a program developed to identify lipid binding surfaces. In the study, the authors describe the impact of different mutations present in the subvariants on the conformation and on the interaction with the binding surface. The results are linked to data reported in the literature, notably on the effects of some mutations on the humoral response (resistance to neutralizing antibodies) and on the affinity for the ACE2 receptor.

Overall, the manuscript is well written, and the results appear solid and are clearly presented. It highlights the differences in structure between the Omicron subvariants, which helps explain the differences in phenotypes observed between them. One of the limitations of the study is that for some variants, few structures are available.

Author Response

We are thankful for the positive comments saying that our manuscript is solid, clearly presented and well written. We agree that there are few structures for some SARS-CoV-2 Omicron subvariant spike ectodomains and have searched the RCSB PDB and Cov3D databases again for more entries (there are none). We clarify that we have analyzed of the available spike trimers and their ACE2 complexes as listed in Table 1 by additional methods, now including DREAMM as well as MODA, PPM3 and the Protein Dipole Moments Server). The results are sufficient to show consistent patterns between the variant spikes and significant correlations, as shown in Figures 2, 3, 4 and 5. We also now state that “Although beyond the scope of this work, the determination of further structures of these spike variants would allow the significance and pathway of membrane interactions to be more fully explored.” We have also improved the paper including providing more detail in the methods, results and discussion to strengthen the revised manuscript, which includes new content in sections 3.4 and 3.5.

Reviewer 2 Report

In the article “Uncompromised membrane binding by SARS-CoV-2 Omicron variants evolved for immune system evasion” the authors investigated the membrane and ACE2 binding propensity of the SARS-CoV-2 Omicron variant, and some of its newer subvariants. Predicting the possible outcomes of the mutations occurring during the evolution of the variants has merits and currently is quite relevant. The methodology only involves computational methods, and the overall presentation of the results is good.

The authors mention immune system / NAb evasion multiple times, and in the Methods, they state that antibody binding sites are also identified. However, reading the article, it is not clear what do they mean, how these are achieved and how they used the Ab binding data.

In the discussion some membrane binding mechanisms are mentioned (e.g. heparin sulphate proteoglycans). The article could benefit from more explanation on how these mechanisms are modeled in the mentioned lipid bilayer slab.

Author Response

We are thankful for the positive comments regarding the merits and relevancy of our manuscript. In the revised manuscript we have included a description of antibody binding, including the additional references in a new section 3.4:

“Common mutations that help COVID-19 strains including Omicron evade the immune system while retaining host receptor binding include K417N, L452R, E484K and F486L in the RBD where most antibodies bind. While these mutations ensure similar or higher ACE2 affinities of the RBD [39], only the latter two substitutions are predicted to directly insert into membranes, while the other two are next to membrane binding motifs. This suggests that enhancing ACE2 binding is more critical to viral success than further optimization of lipid bilayer interactions.

The interactions of antibodies, ACE2 receptors and lipid bilayers can be contrasted to shed light on their respective influences on transmissibility of Omicron sublineages, with BA.2.75 being a particularly competitive outlier. This variant is endowed with a unique antibody avoidance and much higher ACE2 affinity as well as greater fusogenicity, growth efficiency, and intrinsic pathogenicity over BA.2, giving it an edge that may largely be due to acquisition of N460K and R493Q in the RBD [24, 25, 40]. These positions are not predicted to directly bind membrane in the respective spike trimer structures, although they are close to F456 and F490 which are. Hence receptor binding and immune escape can be interpreted to represent primary and secondary drivers for the emergence of such mutations, which maintain host receptor and membrane binding capacities while avoiding antibody interactions.”

We have also explained how the ligands including heparin sulphate proteoglycans are bound in the discussion of the revised version:

“The SARS-CoV-2 RBD interacts with glycolipids [55], in particular those containing sial-ic acid [55] and heparan sulfate (HS) proteoglycans, which enhances ACE2 binding [56]. The RBDs of Omicron sublineages appear to offer differing sites that accommodate HS with nanomolar affinities [57]. By analogy to other spike protein HS complexes [58] we infer that such interactions are responsible for initial docking of the virus to the extra-cellular surface of host cell membranes.”

Reviewer 3 Report

The manuscript described the membrane binding surfaces of Omicron subvariants are compared using cryo-electron microscopy structures of spike trimers by prediction programs. This paper propose that balancing the functional imperatives of cleavable spike proteins in omicrons while avoiding host defenses underlies betacoronavirus evolution.

Major revision

1.      Coronaviruses have been well known for conformational changes in RBD regions due to fast mutation by positive selection. Also, this increased membrane binding capacity of omicrons were already validated in a previous report by this team. (https://doi.org/10.1016/j.isci.2022.104722). Therefore, novelty of this paper seem to be lack.

2.      Membrane binding propensities were predicted by computational analysis using only one programs, MODA. No data and proposal have been experimentally validated by lipid binding assays or structural biology experiment or immune escape experiment by in vitro and in vivo. Experimental data should be needed for proving this proposal.

3.      This predicted data seem not to provide predictive criteria for future pandemic waves and transmissibility.  

Author Response

Authors’ response to comment 1: We feel that although this new manuscript builds on our earlier publication in iScience, the results we present here are novel. The novelty includes the unique analysis of the Omicron subvariants BA.2, BA.2.12.1, BA.2.13, BA.2.75, BA.3, BA.4 and BA.5, none of which was included in the earlier paper. Here for the first time we analyzed 55 higher resolution structures of tSARS-CoV-2 Omicron subvariant spike trimers and their ACE2 complexes using four programs including DREAMM, PPM3, MODA and Protein Dipole Moment Server. This level of analysis of membrane docking has never been reported for any protein by any group to our knowledge. Our manuscript provides novel findings including the data shown in each of the figures, culminating in Figure 5 which shows novel relationships that have not been reported by any other group to our knowledge. Thus we respectfully argue that our manuscript do offer novel findings, consistent with the other viewers who felt that it merited publication.

Authors’ response to comment 2: We used four independent programs in the revised manuscript including the recently published DREAMM program as well as PPM3, MODA and the Protein Dipole Moments Server, the latter of which were used in our original submission. These four methods use different algorithms but identify consistent membrane binding sites based on analysis of the many Omicron spike trimer structures and ACE2 complexes, which have been determined by leading structural biology groups. The experimental data showing direct lipid-spike binding and lipid-bound spike structures is described in our revised submission in section 3.5 as follows:

“There are multiple lipid ligands that interact with spike trimers, as we described earlier [10, 11]. Complexes are formed between spikes and neutral lipids [45], phospholipids [46], fatty acids [34-36], biliverdin [37] and cholesterol [47, 48] that contribute to cell-cell fusion [49] and viral entry [50]. Other candidate ligands include membrane raft components and gangliosides, which can be accommodated within SARS-CoV-2 NTD structures [51, 52] as well as PtdIns(4,5)P2 which colocalizes with spikes in the plasma membrane [53]. Structures of complexes of SARS-CoV-2 spike trimers bound to lipids including fatty acids have been resolved [34, 36, 54] and indicate how cholesterol ligands [47] could influence viral entry [50]. The SARS-CoV-2 RBD interacts with glycolipids [55], in particular those containing sialic acid [55] and heparan sulfate (HS) proteo-glycans, which enhances ACE2 binding [56]. The RBDs of Omicron sublineages appear to offer differing sites that accommodate HS with nanomolar affinities [57]. By analogy to other spike protein HS complexes [58] we infer that such interactions are responsible for initial docking of the virus to the extracellular surface of host cell membranes. Images of native spikes of SARS-CoV-2 viral particles in cells suggest that the heads also abut ER, Golgi, vesicular and plasma membrane surfaces during intracellular assembly and trafficking [59-63], and we propose that similar spike sites mediate lipid inter-actions with such compartments. Altogether this suggests that a diverse array of mem-brane interactions mediated by SARS-CoV-2 virion spike heads throughout their jour-ney into and within host cells are impacted by spike mutations, making the relevant docking surface attractive for intervention.”

Authors’ response:  In the revised manuscript we further clarify and expound on the predictive criteria for transmissibility including a significantly revised section 3.5, and well as an updated title to reflect this increased focus on the predictive criteria:

3.5. Trends and relationships of evolutionary drivers of Omicron spikes

The overall membrane binding propensity of the RBD is retained in Omicron subvariants despite the local variations due to spike mutations. The BA.1 variant spike is particularly mobile, being more prone to repositioning its RBDs, as reflected by the wider range of its MODA scores and dipole moments. In contrast, the membrane binding propensity scores of the BA.2 and BA.2.75 variants are less variable (Fig. 5). The single structure of the BA.2.13 spike exhibits the lowest membrane binding propensity in its RBD while the lowest dipole moment belongs to the BA.2.75 ectodomain. In contrast the BA.2.12.1 and BA.3 spike RBDs both exhibit high membrane binding propensities, with the BA.3 ectodomain displaying the highest dipole moment. Although beyond the scope of this work, the determination of further structures of these spike variants would allow the significance and pathway of membrane interactions to be more fully explored.

Figure 5: Membrane binding by conformational states of SARS-CoV-2 Omicron variants. A) The average membrane binding propensities of the RBD subunit of the closed spike trimer structures are shown for each of the variants. B) The average membrane binding propensities of the subunits of the closed, open, singly, doubly and triply ACE2-bound forms (states 1.0, 1.1, 3.2, 4.2, and 5, Fig. 1) are shown for spike trimer structures (Table 2) of the Omicron BA.2 and BA.2.75 subvariants and C) dipole moments of the closed structures of the Omicron subvariant spike ectodomain trimers, with standard deviations shown unless there are insufficient numbers of structures to calculate this. D) The average dipole moments of the closed structures of the Omicron variant spike ectodomain trimers are linearly correlated with the ACE2 binding affinities of their RBD domains [7, 24, 25].

Interestingly, the dipole moments of the closed Omicron spike ectodomains that attract membranes are correlated with receptor binding. That is, the reported ACE2 affinities [7, 24, 25] exhibit a positive linear relationship based on a linear regression test [F = 14.5, P = 0.0189, R2 = 0.784] (Fig. 5D). This infers that the Omicron variant BA.2.75 has accumulated mutations that increase its ACE2 affinity at the expense of electrostatic attraction to the membrane surface via its reduced dipole moment. Nonetheless, membrane binding propensity does not appear to have been significantly compromised in any of the Omicron variants (Fig 5a). In contrast BA.3 has achieved a strong dipole moment but compromised receptor affinity, while retaining membrane binding propensity. The higher transmissibility of the BA.2.75 subvariant compared to BA.3, which appeared only briefly before being out-competed, suggests that ACE2 affinity outflanks dipole moments, with membrane binding being relatively constant in these Omicron strains. Based on the properties of the mutant spikes discussed here, the drivers for transmissibility of the Omicron variants then include:

  1. Membrane docking propensity via the lipid binding surface of the spike head.
  2. ACE2 receptor binding affinity via the RBM of the open state of the spike trimer.
  3. Neutralization by antibodies involved in immune escape.
  4. Electrostatic membrane attraction via the dipole moment of the spike trimer.

The transmissibility (T) of a SARS-CoV-2 Omicron variant can be predicted as being proportional to the four “MANE” drivers, as described in the following equation:

T ~ 1/wA (xN x yM x zE)

where w, x, y and z represent the relative weightings of the A (ACE2 receptor affinity), N (neutralization of antibodies), M (membrane binding propensity), and D (dipole moment) terms described above. Although transmissibility is dynamic, being dependent on dynamic factors such as vaccination and infection rates and waning immunities, the values of the four driver terms can be measured by receptor binding assays, antibody neutralization data and MODA and dipole moment server analysis of spike structures.

Along with structural data and the tools used here, this provides a basis for predicting the transmissibility of future SARS-CoV-2 variants. A caveat is that transmission rates also depend on other factors such as spike protein stability and protease cleavage, viral entry pathway, replication, assembly and egress. Hence, although the identification of key drivers in past waves can be used to rationalize effects of current variant mutations, future pandemic waves may expose additional determinants of viral fitness.

Round 2

Reviewer 3 Report

The authors have addressed the comments and I recommend acceptance of the manuscript